# MiRNAs as Novel Adipokines: Obesity-Related Circulating MiRNAs Influence Chemosensitivity in Cancer Patients

**DOI:** 10.3390/ncrna6010005

**Published:** 2020-01-23

**Authors:** Sarah B. Withers, Toni Dewhurst, Chloe Hammond, Caroline H. Topham

**Affiliations:** 1Biomedical Research Centre, School of Science, Engineering and Environment, Peel Building, University of Salford, Salford M5 4WT, UK; s.b.withers@salford.ac.uk (S.B.W.); t.j.dewhurst@edu.salford.ac.uk (T.D.); c.j.hammond@edu.salford.ac.uk (C.H.); 2Salford Royal Foundation Trust, Clinical Sciences Building, Stott Lane, Salford M6 8HD, UK

**Keywords:** obesity, microRNA, miRNA, circulating, cancer, chemosensitivity, chemoresistance

## Abstract

Adipose tissue is an endocrine organ, capable of regulating distant physiological processes in other tissues via the release of adipokines into the bloodstream. Recently, circulating adipose-derived microRNAs (miRNAs) have been proposed as a novel class of adipokine, due to their capacity to regulate gene expression in tissues other than fat. Circulating levels of adipokines are known to be altered in obese individuals compared with typical weight individuals and are linked to poorer health outcomes. For example, obese individuals are known to be more prone to the development of some cancers, and less likely to achieve event-free survival following chemotherapy. The purpose of this review was twofold; first to identify circulating miRNAs which are reproducibly altered in obesity, and secondly to identify mechanisms by which these obesity-linked miRNAs might influence the sensitivity of tumors to treatment. We identified 8 candidate circulating miRNAs with altered levels in obese individuals (6 increased, 2 decreased). A second literature review was then performed to investigate if these candidates might have a role in mediating resistance to cancer treatment. All of the circulating miRNAs identified were capable of mediating responses to cancer treatment at the cellular level, and so this review provides novel insights which can be used by future studies which aim to improve obese patient outcomes.

## 1. Introduction

### 1.1. Obesity

The rise of obesity poses a significant global problem. In the UK alone, more than approximately 60% of UK adults are classed as overweight or obese [1]. Concerningly, 41 million children under age 5 are classed as obese and 340 million adolescents are obese [2]. Individuals with a body mass index (BMI) >35 are at an increased risk of mortality independent of age, sex, or race [3]. Obesity is a multifactorial, chronic disease which is associated with a risk of cardiovascular disease [4], cognitive decline [5] and cancer [6]. Current guidelines recommend intervention in individuals who present as overweight and obese [7]. Exercise and dietary programmes produce marginal reductions in risk, whereas surgical intervention is currently reserved for the morbidly obese. Despite this, not all those who have a high BMI will die from conditions associated with obesity, and some of those who present as healthy BMI range, will [8]. Understanding why this happens and whether we can develop better predictive tools is paramount if we are to improve life-expectancy and personalize therapy to patient needs.

Energy imbalance induces an expansion of adipose tissue through hyperplasia and hypertrophy of adipocytes throughout the body. The anatomical location of adipose accumulation is an independent risk factor of disease; visceral accumulation of fat is positively correlated to a number of disorders, including type 2 diabetes [9] heart disease, chronic kidney disease (CKD) [10], breast and colorectal cancer [11,12]. Recent studies have shown that the visceral to subcutaneous fat area ratio (VSR) in central obesity is a poor prognostic factor for oesophageal squamous cell carcinoma and hepatocellular carcinoma [13,14,15]. As such, adipose tissue physiology has gathered research interest over the recent years.

### 1.2. Adipose Tissue

Adipose tissue was traditionally viewed as having a protective role for organs against trauma, storing lipids, and providing thermal insulation [16]. However, its role as an endocrine organ has gathered interest over the past two decades. Adipose tissue is a heterogenous tissue comprised of adipocytes, preadipocytes, immune cells, fibroblasts and stem cells, as well as extracellular matrix proteins [17]. However, adipose tissue depots are diverse in function and type. Adipose tissue has striking heterogeneity, with changes in phenotype relating to its location and physiological function (reviewed [18]). There are two classical types of adipose tissue; brown and white adipose tissue. Brown adipose tissue (BAT) is highly thermogenic and is particularly prominent in infants and hibernating animals [19]. The large number of mitochondria required to produce heat, and the network of blood vessels within BAT are responsible for its colour. It is found around some organs including the heart and kidneys, and larger blood vessels, such as the aorta. Research exploiting its energy expending role is of interest in the fields of obesity and type 2 diabetes [20,21,22,23].

White adipose tissue (WAT) is found throughout the human body surrounding around almost all blood vessels, organs, and muscles [24]. It plays a vital role in energy storage, including the release of free fatty acids when required [16], and its expansion is linked to caloric excess. WAT plays a dynamic role in the regulation of metabolism, inflammation and other physiological processes through the secretion of locally and systemically acting signaling molecules, termed adipokines [25,26].

### 1.3. Adipokines

Adipokines are signaling molecules which act in a paracrine and endocrine manner and can elicit pro- and anti-inflammatory effects. Adipokines influence numerous physiological functions including electrical conductivity in the heart [27], glucose and lipid metabolism [28], blood pressure control [29], cell adhesion, vascular growth and function [30], adipogenesis [31] and bone morphogenesis [32], regulation of appetite, satiety and immune response, and other biological processes [7,33]. We and others have shown, that adipokines are able to mediate the cross-talk between adipose tissue, immune cells, muscles, and the sympathetic nervous system to elicit physiological changes [34,35].

Adipokine research is an expanding field. Much of the published literature has examined how obesity impacts the adipokine profile as a means to understanding the pathophysiological changes linked to high calorie intake. Leptin, resistin, and visfatin are known to be positively correlated with body mass index (BMI) and these pro-inflammatory adipokines have been implicated in regulation of food intake, vascular inflammation, oxidative stress, and vascular smooth muscle hypertrophy [36,37,38], linking obesity to type 2 diabetes and cardiovascular disease. Adiponectin and omentin are examples of adipokines which are negatively correlated with BMI; they have suppressive effects on oxidative stress [39,40,41] and demonstrate some anticancer properties [39,42].

Interestingly, many of these adipokines have been investigated as biomarkers for predicting the risk of disease in obese patients. While the significance of changes in individual adipokines due to obesity remains questionable, the low-grade inflammation associated with adipose tissue expansion is likely to elicit changes in many pathways which can result in pathophysiological effects. For example, the activation state of the transcription factor, nuclear factor kappa-B (NF-κΒ) is linked to adiponectin [43] and resistin [44], among others [45]. Further to this, NF-κΒ has been implicated in a number of conditions including cognitive decline, asthma [46], a number of cancers and chemotherapy response. As such, manipulating adipose-secreted factors offers opportunity to develop new treatment strategies [47]; yet the therapeutic potential of these factors has not yet come to fruition and the complexity of adipose tissue is still increasing.

### 1.4. MicroRNA as an Adipokine

MicroRNAs (miRNAs) are endogenous RNAs of 19–22 base pairs and found in the circulation, as well as in tissues; a large proportion of which are located in exosomes [48]. MicroRNAs are implicated in direct regulation of the expression of >30% human genes [49] through complementary binding at the 3′ untranslated region (3′ UTR) and subsequent inhibition of translation [50]. The enzyme Dicer is crucial for processing of the pre-miRNA to a mature miRNA capable of gene repression [51]. Recent work used adipose tissue-specific knockout of Dicer to demonstrate that adipose tissue is a major source of circulating exosomal miRNAs in mice [48]. The authors also provided evidence that these adipose-derived exosomal miRNAs were able to regulate gene expression in distant tissues such as the liver, and so could be considered a novel class of adipokine. The importance of circulating miRNAs as novel adipokines is still to be clarified, and the work presented here aims to identify relevant avenues of investigation.

Akin to ‘traditional’ adipokines, the levels of miRNAs are associated with a number of different conditions including diabetes [52,53,54], obesity [55], cancer [56], and cardiovascular disease [57]. They have been shown to regulate a plethora of physiological functions at paracrine and endocrine levels; miRNAs have been shown to play a role in adipocyte fat deposition, differentiation and brown adipogenesis [58,59] and circulating adipocyte-derived exosomal miRNAs may regulate whole-body metabolism and mRNA translation in other tissues [48]. In line with adipokines, adipose-miRNAs have been implicated as modulators of the immune response via immune cell and cytokine changes, demonstrating a further mechanism of crosstalk between adipose tissue and inflammation [60].

### 1.5. Adipose Tissue and Its Role in Chemotherapy Resistance

Adipose tissue expansion has also been linked to chemotherapeutic drug resistance [33]. There are a number of mechanisms which have been proposed, divided loosely into drug metabolism or cellular mechanisms [33,61]. Drug metabolism changes due to adiposity include altered pharmacokinetics, changes in drug delivery, and lipid loaded adipocytes acting as energy stores [62]. However, changes in adipokine secretion have also been linked to resistance; leptin, which is increased in obesity, stimulates proliferation of multiple myeloid cells and activates AKT, which is linked to reduced anti-tumor capacity of chemotherapy [63]. Further to this are the changes in cell signaling pathways brought about by low-grade inflammation linked with obesity; pro-inflammatory adipokines such as IL6 and TNF-α promote tumor progression and survival (reviewed [64]). Many more associations have been documented, but in the context of this review, this serves to highlight the potential of adipose-secreted factors to impact on treatment success in cancer and therefore to consider whether adipose-secreted miRNAs can act similarly.

The purpose of this review was to collate current knowledge on circulating miRNAs in obese and typical weight individuals, and to link this understanding to the growing knowledgebase of how miRNAs can affect sensitivity to therapeutics in cancer cells. Adipose-derived miRNAs can act as adipokines by regulating gene expression in distant tissues. When combined with the knowledge that some obese patients have poorer responses to anti-cancer treatments [65,66], these observations prompted us to explore how circulating miRNAs which are altered in obesity might influence treatment sensitivity in distant tumor cells

## 2. Results

We have performed a comprehensive literature search to identify circulating miRNAs which are up or down-regulated in obese individuals, compared with typical weight individuals. This stage 1 search builds upon the recent comprehensive review of Zaiou et al. [67] and identified 67 differentially regulated miRNAs, 34 that were increased in obesity and 33 that were decreased in obesity (see Appendix A, Table A1). After exclusions were applied as described in Methods, eight miRNAs were taken forward to stage 2, six of which were increased in obese individuals (Table 1), and two of which were decreased (Table 2).

## 3. Discussion

Recent work identified miRNAs as novel adipokines, capable of exerting phenotypic effects in distant tissues [48]. This proposed role for circulating miRNAs is relatively unexplored, so we have performed a comprehensive literature review to identify circulating miRNAs which are altered in obesity in order to identify putative adipokine miRNAs relevant to cancer treatment success. Six miRNAs were found to be reproducibly increased in the circulation of obese individuals: miRNA-122, miRNA-140-5p, miRNA-142-3p, miRNA-143, miRNA-222, and miRNA-486, and two miRNAs were found to be decreased: miRNA-221 and miRNA-520c-3p.

We have taken a reductionist approach to investigate how altered circulating expression levels of these miRNAs could influence cancer treatment success, by reviewing the literature available regarding each miRNA individually and its role in sensitivity to cancer treatment, a summary of which is provided below. In reality, miRNAs do not operate in isolation, and it is likely that understanding the wider networks of these miRNAs and their targets is required to identify clinically meaningful patterns. This review serves as a starting point for investigating these key networks.

This work extrapolates from the obesity-linked changes in circulating miRNA levels to the predicted impact these changes may have on the success of standard treatment regimens. A limitation of this work is these predictions rely on the assumption that circulating miRNAs can reach and influence the tumor of interest, however it is feasible this knowledge could then be used to identify potentially meaningful relationships, with a view to designing better treatment regimens for obese cancer patients.

### 3.1. Circulating Levels of MiRNA-486 Are Increased in Obesity

Circulating miRNA-486 levels are increased in obese individuals, specifically children (Table 1). The role of miRNA-486 in cancer has been widely studied and increased levels have been found in prostate cancer [76], glioma [77], non-small-cell lung cancer (NSCLC; [78]) and chronic myeloid leukemia (CML; [79]). Conversely, its down-regulation has been reported in cancers including those of the cervix and breast [80,81], osteosarcoma [82] and, contradicting the oncogenic role described above, also in NSCLC [79]. This suggests a complex role for miRNA-486, which is known to repress targets such as members of the PI3K/AKT signaling pathways PTEN [83] and FoxO1 [79], and the metastatic regulator Snail [84]. Therefore miRNA-486 has a clear role in tumor progression, although whether this is an oncogenic or tumor suppressive role appears to be tissue-specific, as will be discussed in the following sections.

#### 3.1.1. Increased MiRNA-486 Levels Enhance Treatment Sensitivity in Non-Small Cell Lung and Breast Cancer

Pim-1, a proto-oncogenic kinase, was increased in NSCLC tissue samples and found to be a target of miRNA-486 in this tumor type [78]. Consistent with this, miRNA-486-5p expression levels were consistently decreased in these NSCLC tissues. Reduction of Pim-1 using siRNA was able to sensitise NSCLC cell lines to both cisplatin and the EGFR tyrosine kinase inhibitor (TKI) gefitinib. Although the effect of over-expressing miRNA-486 on drug-sensitivity in these cell lines was not tested in this study, this correlative evidence suggests that increasing levels of miRNA-486 could promote sensitivity to cisplatin and gefitinib in NSCLC.

In another study, MCF7 breast cancer cells resistant to either fulvestrant (MCF7-F) or tamoxifen (MCF7-T) were generated, and miRNA expression profiles generated using microarray and compared with drug-sensitive parental MCF7 cells. Fourteen miRNAs were found to be down-regulated in both drug-resistant cell lines, including miRNA-486. No further work was done to investigate the functional effect of this down-regulation, but this raises the possibility that increasing levels of miRNA-486 in ER+ breast cancer may restore sensitivity to treatment [85].

#### 3.1.2. Increased MiRNA-486 Levels Promote Resistance to Treatment in Lung Adenocarcinoma, Chronic Myeloid Leukaemia and Colorectal Cancer

In contrast to the previous section, studies also exist which describe the role of miRNA-486 in chemo-resistance. Extracellular vesicles (EVs) produced by *ALK*-translocated lung adenocarcinoma cell lines, which were rendered resistant to the ALK-TKIs crizotinib or ceretinib, were found to contain elevated levels of miRNA-486 amongst other non-coding RNAs [86]. When applied to drug-sensitive cell lines, the miRNA-486-containing EVs conferred resistance to both TKIs.

Evidence also suggests a role for miRNA-486 in the sensitivity of blood cancers to treatment, specifically chronic myeloid leukemia (CML). In addition to identifying a role for miRNA-486 in normal erythropoiesis, its repression via transduction of an anti-miRNA-486-5p vector into BCR-ABL-expressing cells resulted in increased apoptosis, which was further enhanced when combined with imatinib treatment [79]. So by reducing miRNA-486 levels, increased sensitivity to imatinib was restored, due to the relief of repression of miRNA-486 targets PTEN and FoxO1 and concurrent reduction in phosphorylation of Akt.

Another study [87] provides further evidence of the capacity of miRNA-486 to influence treatment outcomes, showing that colorectal cancer (CRC) patients treated with a vaccine using 5 epitope peptides were more likely to have an improved prognosis if the tumors exhibited low miRNA-486 expression levels.

#### 3.1.3. Circulating MiRNA-486 expression may influences treatment sensitivity in a tissue-dependent manner

It is clear that the influence of miRNA-486 levels in response to treatment appears to be tissue-dependent, with potential for NSCLC and breast tumors to be sensitised to a range of treatments, with the caveat that *ALK*-translocated NSCLC experience the opposite effect, becoming more resistant to treatment. CML and CRC became more resistant to treatments when expression levels of miRNA-486 were high. Most of these cancers are not relevant to pediatric populations, and whether or not elevated circulating levels of miRNA-486 are elevated in adults as well as children remains to be seen, but the evidence summarized here strongly suggests that this miRNA is capable of influencing treatment responses. In conclusion, the levels of circulating miRNA-486 in adult obese patients should be explored, as it may be a biomarker for poorer treatment success in *ALK*-translocated NSCLC, CML or CRC. In the case of obese pediatric patients, increased levels of circulating miRNA-486 may be a biomarker for poorer outcomes in CML, and the impact of increased levels of this miRNA in other pediatric tumor types should be investigated.

### 3.2. Circulating Levels of the Related Paralogs MiRNA-221 and MiRNA-222 Are Altered in Obesity

Circulating miRNA-222 levels were found to be increased (Table 1) and miRNA-221 levels were decreased (Table 2) in obese individuals. This was unexpected as the closely related miRNAs-221 and -222 are located together on the X chromosome and are generally expressed as a cluster [88]. Both miRNAs have previously been linked to therapeutic responses in cancer [88]. They also share an identical seed sequence so are expected to regulate a shared pool of target genes [88]. This raises an interesting paradox for this literature review as circulating miRNA-222 levels were identified as elevated in obese individuals yet circulating miRNA-221 levels were decreased. According to our criteria, five studies identified obesity-associated increased miRNA-222 levels (Table 1), however only two studies described reduced circulating miRNA-221; one performed using pediatric samples [68] and the other in adult males [71]. On closer inspection, the study by Ortega et al. saw a non-statistically significant increase in circulating miRNA-221 when comparing lean with obese individuals, and it was only in the morbidly obese that levels dropped. Therefore, it is possible that both miRNA-221 and -222 are generally increased in the circulation of obese adults, and that miRNA-221 levels are decreased only in morbidly obese adults and obese children. Both miRNAs were found to be upregulated in subcutaneous adipose tissue of obese individuals [67] but miRNA-221 was found to be down-regulated in visceral fat tissue of obese individuals. Both miRNAs are dysregulated in several cancers and the following sections review their role in the sensitivity of these cancers to treatment.

#### 3.2.1. MiRNA-221 and -222 Influence Treatment Sensitivity in Glioblastoma

There are conflicting reports regarding the role of increased levels of miRNA-221/222 on treatment sensitivity in glioblastoma. Standard chemotherapy for glioblastoma includes temozolomide, and cytotoxicity is enhanced in T98G glioblastoma cells in the presence of a miRNA-222 mimic and reduced in the presence of a miRNA-222 inhibitor [89]. The authors also present data which suggest this enhanced sensitivity to treatment in the presence of increased levels of miRNA-222 is due to repression of the long non-coding RNA GAS5 and the DNA damage repair protein MGMT. This study is supported by another reporting that increased expression of miRNA-221/222 correlated with enhanced DNA damage and apoptosis in glioblastoma multiforme in response to temozolomide, due to repression of the miRNA-221/222 target, the DNA repair protein MGMT [90].

There is also evidence to the contrary. Peptide nucleic acids (PNAs) targeting either or both miRNA-221 or -222, resulting in a decrease of their levels, were found to be cytotoxic to three glioma cell lines [91]. The temozolomide-resistant cell line, T98G, was re-sensitised to this treatment in the presence of either or both of these PNAs.

In glioma, increased miRNA-221 is positively correlated with tumor grade [76]. Using glioma cell lines, the authors were able to demonstrate that inhibiting miRNA-221 was able to enhance sensitivity of glioma cells to temozolomide, via relief of suppression of DNM3. Inhibition of miRNA-221 can also increase sensitivity to carmustine in glioma cells via upregulation of PTEN, further supporting the oncogenic role of this miRNA in gliomas [92].

Increased expression of miRNA-221/222 is known to occur in glioblastoma [93] and glioma [92] and the majority of the literature discussed here suggests this contributes to chemo-resistance. There is more evidence that miRNA-221 is responsible for this than miRNA-222 but further studies will clarify this.

#### 3.2.2. Increased MiRNA-221/222 Levels Promote Resistance to Treatment in Breast Cancer

Increased expression levels of miRNA-222, amongst several other miRNAs, has also been linked with treatment resistance in formalin-fixed breast tumor samples [94], as levels were elevated in post-chemotherapy samples when compared with pre-chemotherapy biopsies. This pattern holds true for several breast cancer treatments, as outlined below.

##### Tamoxifen

The GAS5-miRNA-222 axis can also influence sensitivity to tamoxifen in breast cancer. In this context, miRNA-222 mimics were shown to promote growth of tamoxifen-resistant MCF7 cells in relation to control populations [95]. The repression of PTEN by miRNA-222 may be responsible for this enhanced resistance to treatment [95]. Another study used antisense oligonucleotides targeting miRNA-221 and/or miRNA-222 to enhance the sensitivity of ER+ MCF-7 cells to tamoxifen via the relief of suppression of the miRNA-221/222 target TIMP3 [96]. Increased levels of miRNA-222 in tamoxifen-resistant MCF7 cells can arise due to down-regulation of the long noncoding RNA GAS5, which sequesters miRNA-222 by acting as a “molecular sponge” [95].

##### Adriamycin and Docetaxol

Also linked to miRNA-222 mediated chemoresistance in breast cancer cells is the PTEN/Akt/FOX01 pathway, as adriamycin-resistant MCF-7 cells exhibited elevated levels of miRNA-222 [97]. The authors were able to demonstrate a link with the PTEN pathway using miRNA mimics and inhibitors. They also reported poorer overall survival outcomes for 729 breast cancer patients whose tumors expressed higher levels of miRNA-222 [97].

In agreement with the above observations, miRNA-222 mimics can promote resistance to adriamycin and docetaxol in MCF-7 breast cancer cells, with PTEN identified as the putative target of miRNA-222 [98]. Interestingly, exosomes produced by drug-resistant MCF-7 cells can deliver miRNA-222 to previously sensitive lines, rendering them resistant to adriamycin also [63]. Conversely, miRNA-222 inhibitors enhance the sensitivity to adriamycin and docetaxel [94].

##### Fulvestrant

MiRNA-221 and -222 were found to be over-expressed in fulvestrant-resistant MCF-7 breast cancer cells compared to sensitive lines, and were also required for proliferation in these resistant lines [99]. Expression of several genes were altered after ectopic expression of miRNA-221/222 in MCF-7 cells, including a reduction in p27^KIP1^ mRNA levels [99], a CDK inhibitor and known target which has also been shown to mediate miRNA-222-induced tamoxifen resistance in MCF7 cells [100].

##### Cisplatin

Elevated miRNA-221 levels are linked to resistance to cisplatin in the triple-negative model MDA-MB-231, with suppression of miRNA-221 leading to increased sensitivity to cisplatin. This was proposed to be mediated by relief of suppression of the miRNA-221 target Bim, a pro-apoptotic member of the Bcl-2 family [81].

##### Trastuzumab

Sensitivity to trastuzumab in HER2+ breast cancer cells is also influenced by miRNA-221, probably via the actions of PTEN as overexpression of this miRNA-221 target was able to rescue sensitivity to trastuzumab in cells over-expressing miRNA-221 [81].

The above evidence clearly demonstrates that high levels of miRNA-221 and or miRNA-222 may be a poor prognostic factor in the treatment of breast cancer by conferring resistance to hormone therapy, targeted therapy, and cytotoxic chemotherapy.

#### 3.2.3. Increased MiRNA-221/222 Levels Promote Resistance to Treatment in Non-Small Cell Lung Cancer

Following the same pattern observed with breast cancer, miRNA-221/222 have a clear role in increasing resistance to treatment in NSCLC. In NSCLC cells suppression of Dicer was found to be a sensitiser to gefitinib, highlighting the importance of miRNAs to chemotherapy responses in this tumor type. The key downregulated miRNAs in this context were identified to be miRNA-221/222 and miRNA-30b/c which target CASP3 mRNA, so elevated caspase 3 levels in dicer-depleted NSCLC cells was identified as the mechanism of sensitization to gefitinib [94].

Both miRNA-221 and -222 can regulate sensitivity to gefitinib in NSCLC via repression of apoptotic peptidase activating factor 1 (APAF-1), downstream of the EGF and MET receptors [88]. Overexpression of these two miRNAs in cell lines sensitive to gefitinib was sufficient to reduce the sensitivity to this treatment, and inhibition of miRNA-221/222 had the opposite effect. Repression of miRNA-221 also resulted in increased gefitinib-induced apoptosis in an A549 cell xenograft mouse model. Another study investigating mechanisms of resistance to TRAIL in HCC and NSCLC identified that miRNAs-221 and -222 were increased in cell lines with lower sensitivity to TRAIL, compared with more sensitive lines. These miRNAs were found to be regulated downstream of the *MET* oncogene to mediate this resistance via repression of PTEN and TIMP3 [88].

Exosomes released from a gemcitabine-resistant NSCLC cell line, A549-GR, were able to confer resistance to this treatment to previously sensitive A549 cells [101]. Several miRNAs were found to be upregulated in A549-GR exosomes, including miRNA-222-3p which was concluded to be a key regulator of the drug-resistance phenotype via inhibition of its target SOCS3, a negative regulator of the JAK-STAT pathway. Interestingly the authors also investigated the levels of exosomal miRNA-221-3p in patient sera and found elevated levels in patients with a poorer response to gemcitabine treatment, plus an increased chance of developing metastases. Other miRNAs found to be increased in exosomes of gemcitabine-resistant cells include miRNA-143-3p [101] which is also of interest to this review due to its elevated circulating levels in obese individuals (Table 1 and Section 3.7). In summary, elevated levels of miRNAs 221/222 can promote resistance to treatment in NSCLC.

#### 3.2.4. Increased MiRNA-221 Levels Promote Resistance to Treatment in Pancreatic Cancer

In pancreatic cancer cell lines miRNA-221-3p overexpression correlates with reduced sensitivity to 5-fluorouracil and gemcitabine and increased migration, possibly via inhibition of RB1 [102]. Treatment with lapatinib and capecitabine of pancreatic cancer cell lines with intrinsic resistance to these treatments resulted in increased expression of miRNA-221 and -210, when compared with more sensitive cell lines [103]. Inhibition of miRNA-221 was able to sensitise PANC-1 cells to treatment, suggesting a role for this miRNA in mediating chemo-resistance in pancreatic cancer.

As previously discussed, the lncRNA GAS5 also targets miRNA-221 and can reduce its levels in pancreatic cancer cells resulting in increased sensitivity to gemcitabine via relief of suppression of the miRNA-221 target SOCS3 [104]. Also, in pancreatic cancer, metformin was able to suppress miRNA-221, leading to elevated levels of Bim and p27 and sensitizing p53-mutant cells to TRAIL [105]. Our literature search did not identify miRNA-222 as related to treatment responses in pancreatic cancer, but it is clear that elevated miRNA-221 can promote resistance to treatment in this tumor type.

#### 3.2.5. MiRNA-221 and -222 Influence Treatment Sensitivity in Colorectal Cancer

Contrasting evidence exists regarding the role for miRNA-222 to treatment resistance of colorectal cancer cells (CRC). One study found that CRC cells with resistance to vincristine or oxaliplatin expressed lower levels of both miRNA-222 and -221, when compared with drug-sensitive parental lines. Mimics of miRNA-222 were able to enhance sensitivity to treatment, via repression of the miRNA-222 target ADAM-17 [106].

However, a later study found miRNA-222-3p to be upregulated in doxorubicin-resistant LoVo cells. Sensitivity to treatment was restored by inhibition of this miRNA and siRNA knockdown of FOXP2 reverted the cells back to a drug-resistant phenotype, suggesting that FOXP2 is the key target of miRNA-222 involved in rendering CRC resistant to doxorubicin [74]. More studies are required to clarify the role of these miRNAs in therapeutic response in CRC.

#### 3.2.6. MiRNA-221 and -222 Influence Treatment Sensitivity in Leukaemia

Interestingly in blood cancers, miRNA-221 and miRNA-222 seem to have opposing functions, depending on the type of leukemia. In MLL-AF4 rearranged acute lymphoblastic leukemia (ALL) cell lines, overexpression of miRNA-221 was able to sensitise to dexamethasone treatment, an effect that was greatly amplified by simultaneously overexpressing miRNA-128b [107].

In chronic myeloid leukaemia (CML) cell lines, increased levels of miRNA-221 were also found to enhance response to treatment, in this case restoring sensitivity to imatinib in the resistant cell line K562/G via repression of STAT5 [108]. A reduction in expression of miRNA-221 was also measured in peripheral blood mononuclear cells (PBMC) from patients with treatment failure when compared with patients with optimal responses [108].

However, in chronic lymphocytic leukemia (CLL) patients with acquired resistance to fludarabine, miRNA-222 was found to be expressed at elevated levels, along with miRNA-21. When antisense oligonucleotides were used to inhibit these miRNAs, enhanced sensitivity to fludarabine was observed in MEG-01 cells [109].

Some blood cancers could be particularly sensitive to circulating miRNAs, as they are in direct contact, and it will be of interest to see if these miRNA expression patterns correlate with treatment success.

#### 3.2.7. MiRNA-221 and -222 Influence Treatment Sensitivity in Oral Squamous Cell Carcinoma

Similar to leukemias, miRNA-221 and -222 seems to have opposing functions in oral squamous cell carcinoma (OSCC). MiRNA-221 is elevated in response to doxorubicin (dox) treatment in OSCC and is associated with insensitivity to this treatment [110]. Inhibition of miRNA-221 using antisense oligonucleotides was able to increase apoptosis in response to dox, via the relief of suppression of the miRNA-221 target TIMP3. In contrast miRNA-222 mimics can promote sensitivity to cisplatin in tongue squamous cell carcinoma by reducing expression of ABCG2 [78].

#### 3.2.8. MiRNA-221 Enhances Treatment Sensitivity in Cholangiocarcinoma

The cholangiocarcinoma (bile duct cancer) cell line, HuH28, was sensitised to gemcitabine in the presence of mimics of miRNA-221, miRNA-125 and miRNA-29b [111]. The key target of miRNA-221 in this context was PIK3R1, and siRNAs targeting this mRNA was able to mimic the sensitisation to gemcitabine seen with the miRNA-221 mimic [111].

#### 3.2.9. MiRNA-221 and -222 Enhance Resistance to Treatment in Other Cancers

##### Bladder

Treatment sensitivity in bladder cancer is reported to be regulated by miRNA-222. Decreased sensitivity to cisplatin was observed in several cell lines following transfection with miRNA-222 mimics, and could be rescued by expression of exogenous PPP2R2A, a known target of miRNA-222 [112]. The repression of PPP2R2A resulted in activation of Akt/mTOR signaling and so the authors conclude that the decrease in cisplatin sensitivity mediated by miRNA-222 overexpression was due to decreased autophagy [112].

##### Multiple Myeloma (MM)

Here, miRNA-221/222 levels inversely correlated with sensitivity to dexamethasone (dex). The mode of death in these cell lines in response to dex was found to be autophagic, regulated via the novel miRNA-221/222 target ATG12 upstream of the p27-mTOR signaling pathway [113]. Also, in MM, high miRNA-221/222 was linked with low sensitivity to melphalan, and inhibitors of these miRNAs were able to restore sensitivity to this treatment in both MM cell lines and melphalan-refractory MM xenografts, in part due to modulation of the miRNA-221/222 target PUMA [114].

##### Hepatocellular Carcinoma (HCC)

In models of HCC, inhibitors of miRNA-221 were able to promote sensitivity to TRAIL in TRAIL-resistant cells [115], an observation confirmed by Jin et al. [116] who also identified that the lncRNA CASC2 was able to sequester miRNA-221, along with miRNA-24, in order to mediate the response to TRAIL treatment. Increased miRNA-221 expression has also been linked to sorafenib-resistance in two mouse models of HCC, with caspase 3 as the downstream target of miRNA-221 able to influence chemosensitivity [117]. Therefore miRNA-221 has been proposed as a biomarker of treatment sensitivity in HCC patients [117].

##### Osteosarcoma

Upregulation of miRNA-221 results in cisplatin resistance, and its inhibition resulted in increased apoptosis in both untreated and cisplatin-treated cells [102]. PTEN was identified as the key target in this context, and overexpression of PTEN could rescue cisplatin-induced cell death in cells with high miRNA-221 levels [102].

#### 3.2.10. Elevated Levels of Circulating MiRNA-222 May Promote Resistance to Cancer Treatment in Obese Individuals

From the extensive evidence above, miRNAs-221 and -222 have clear role in promoting resistance to therapy, as has previously been described (reviewed [88]). There are some contrasting data, with published examples of tumor sensitization in the presence of elevated miRNA-221/222 levels in the case of glioblastoma, leukemias and cholangiocarcinoma and CRC, but the bulk of evidence clearly supports a role in chemo-resistance, particularly in tumors of the breast, lung, and pancreas. A recent meta-analysis confirmed that high expression of miRNA-222 is associated with a poor prognosis in cancer patients although the significance of miRNA-221 was less clear [118]. Increased levels of miRNA-221 are linked with treatment resistance in glioblastoma, breast cancer, NSCLC, pancreatic cancer, MM, HCC, and osteosarcoma; however, we did not establish that levels of miRNA-221 are clearly elevated in the circulation of obese patients. Our literature review did identify elevated levels of circulating miRNA-222 in obese individuals, so we conclude that obese patients may have an enhanced risk of resistance to treatment for breast cancer, NSCLC bladder cancer and MM.

### 3.3. Circulating Levels of MiRNA-122 Are Increased in Obesity

MiRNA-122 is found to be enriched in the liver and accounts for between 50% and 70% of its total miRNA population [119]. A pro-adipogenic role for miRNA-122 has been proposed, and circulating levels are found to be increased in obesity, though there is some contention as to levels in the morbidly obese. Further to this, age and race may impact on the relationship between BMI and miRNA-122 levels [71]. Despite the risk of hepatocellular carcinoma (HCC) in patients with chronic hepatitis C increasing with BMI [120,121], miRNA-122 down-regulation is a common feature of HCC in both human and mice [117] and low miRNA-122 levels correlate with poor prognosis [122].

MiRNA-122 has been found to target a number of genes linked with tumour progression, including Snail 1 and Snail 2, WNT1, CREB1, and BCL9 [123]. Overexpression of miRNA-122 in cell lines reduces their tumorigenic capacity and enhances sensitivity to chemotherapy. Targeting miRNA-122 may be a promising strategy for HCC where there is significant resistance to chemotherapy. Overexpression of miRNA-122 in cell lines reduces their tumorigenic capacity and enhances sensitivity to chemotherapy. Targeting miRNA-122 may be a promising strategy for HCC where there is significant resistance to chemotherapy [124,125].

#### 3.3.1. MiRNA-122 Enhances Treatment Sensitivity in Hepatocellular Carcinoma

As a liver specific miRNA, the role of miRNA-122 in HCC has been extensively investigated. In HepG2 cells overexpressing miRNA-122, sensitivity to doxorubicin was enhanced due to repression of cyclin G which resulted in p53 activation and apoptosis [117]. Consistent with this finding, HCC patients with low miRNA-122-expressing tumors experienced relapse sooner than patients with high levels of miRNA-122, as did patients with high cyclin G-expressing tumors, further supporting the inverse correlation between the two. Overexpression of miRNA-122 can also sensitise HCC cells to adriamycin or vincristine [126]. Doxorubicin-resistant HCC cell lines also display reduced miRNA-122 expression, and this resistance can be reversed by overexpression of miRNA-122 [127]. In this context the miRNA-122 target and glycolytic enzyme PKM2 had a dominant role in dictating chemosensitivity, as overexpression of PKM2 could rescue doxorubicin-resistance in the presence of a miRNA-122 mimic. PKM2 also plays a role in mediating resistance to 5-FU in HCC cells due to low expression of miRNA-122, and restoration of miRNA-122 levels sensitises cells to this treatment and reduces PKM2 levels accordingly [82].

The role of miRNA-122 as a treatment-sensitiser in HCC has also been observed in response to targeted therapy, specifically the multi-kinase inhibitor sorafenib [128]. ADAM10, serum response factor (SRF) and the tyrosine kinase receptor IGF-1R were validated as the miRNA-122 targets relevant to response to sorafenib in this study. This is supported by work by Xu et al. [129] who describe reduced expression of this miRNA in sorafenib-resistant cells. This study confirms that low levels of miRNA-122 result in elevated expression of its target IGF-1R, leading to activation of Ras/Raf/Erk signaling despite the presence of sorafenib. They also show convincing evidence that miRNA-122 levels are low and IGF-1R levels are high in HCC tissue samples from sorafenib-resistant patients [129]. The protease inhibitor SerpinB3 is another target of miRNA-122 relevant to its role in mediating sorafenib-resistance in HCC [130].

One interesting study by Lou et al. [124] described the direct effect of adipose-derived exosomes on chemosensitivity of HCC. MiRNA-122 levels were elevated in adipose-derived mesenchymal stem cells using an overexpression vector, and exosomes from these cultures harvested and applied to the HCC cell line HepG2, resulting in increased sensitivity to 5-fluororacil and sorafenib. They also observed reduced tumor growth in a HepG2 xenograft mouse model treated with sorafenib [124]. This study is particularly interesting as it supports the hypothesis that exosomes released from adipose tissue in obese patients can directly alter sensitivity to chemotherapy in other organs, such as the liver. One study challenges the role of miRNA-122 as a sensitiser to treatment in HCC cells and describes elevated miRNA-122 levels as able to confer resistance to 5-FU in HCC, via regulation of PCDH20 which acts upstream of Akt signaling pathways [131].

The weight of evidence reviewed here strongly supports the role of miRNA-122 as a sensitiser to multiple cytotoxic and targeted treatments in HCC. However, our systematic literature review identified circulating miRNA-122 at higher levels in obese patients compared with typical weight individuals, which would suggest obese individuals may be more responsive to treatment for HCC. This contradicts the widely reported role of obesity as a pathogenic promoter of liver cancer (reviewed in [132]) and the increased chance of HCC relapse observed in obese patients [133]. Therefore, it appears the levels of circulating miRNA-122 in obese patients in unlikely to offer useful prognostic information regarding HCC treatment.

#### 3.3.2. MiRNA-122 Influences Treatment Sensitivity in Colorectal Cancer and Lymphoma

Elevated miRNA-122 levels may promote resistance to treatment in cutaneous T-cell lymphoma (CTCL), as miRNA-122 mimics were shown to promote resistance to γ-secretase inhibitors (GSIs), and the proteasome inhibitors bortezomib and MG132 in these diseases, via inhibition of p53 and activation of Akt [134]. Inhibition of this miRNA sensitises cells to these treatments and p53 was identified as an upstream regulator of miRNA-122 in this context.

In CRC the anti-apoptotic protein XIAP was identified as a target of miRNA-122 and was elevated in oxaliplatin-resistant cell lines compared with sensitive lines, with concurrently low miRNA-122 levels. Restoring miRNA-122 levels in these cell lines resulted in increased sensitivity to oxaliplatin [135].

### 3.4. Circulating Levels of MiRNA-142 Are Increased in Obesity

MiRNA-142 has an emerging role as a regulator of development and has been described as both a tumor suppressor and tumor promoter in several cancer types (reviewed in [136]). This literature review identified miRNA-142 as elevated in the circulation of obese individuals, and the available literature describing the role of this miRNA in the cellular response to cancer treatment is summarized in the following sections.

#### Increased MiRNA-142 Levels Enhance Treatment Sensitivity in Cancers of the Pancreas, Ovary, Lung, Liver, and Blood

This literature review identified several articles describing a role for miRNA-142 in sensitizing several different cancer types to treatment. Pancreatic cancers were sensitised to adriamycin treatment in response to elevated miRNA-142 expression [137]. The relevant miRNA-122 target was DJ-1, an antagonist of PTEN, so elevated miRNA-142 was able to repress DJ-1 resulting in relief of suppression of PTEN and inhibition of Akt signaling.

Increased sensitivity to chemotherapy when miRNA-142-3p is overexpressed has also been observed in HCC, specifically in response to 5-Fluorouracil (5-FU) and cisplatin [138]. Interestingly the mechanism identified here was inhibition of CD133, a stemness marker and direct target of miRNA-142, resulting in a reduction in cancer stem cell-like properties in HCC cells. HCC patient specimens with high miRNA-142 levels were found to have longer disease-free survival rates. Like miRNA-122, miRNA-142 can sensitise HCC cells to sorafenib and another study identified that inhibition of autophagy via repression of the autophagy related proteins ATG5 and ATG16L1 was the relevant mechanism [139].

Acute myelogenous leukemia (AML) cells were also sensitised to treatment when miRNA-142-3p levels are increased, via repression of high mobility box group 1 (HMGB1), an autophagic promoter, and P-glycoprotein, a drug efflux pump [140]. AML cell lines resistant to either all-trans retinoic acid or adriamycin were stimulated to undergo apoptosis when transfected with miRNA-142-3p alongside the relevant treatment. It is worth noting that transection of miRNA-142-3p alone also resulted in some cytotoxicity in the absence of drug.

The importance of relief of suppression of the miRNA-142 target HMGB1 in mediating enhanced sensitivity to treatment when levels of this miRNA are increased is also supported by observations in NSCLC cell lines [141]. This study described the inhibition of HMBG1-mediated autophagy in response to increased expression of miRNA-142, and the resulting enhanced sensitivity to Adriamycin and cisplatin.

In ovarian cancer, miRNA-142 has been shown to sensitise cell lines to cisplatin by repressing XIAP, a mediator of chemoresistance in this tumor type, and other apoptotic regulators [142]. In patient samples, expression of XIAP was inversely correlated with miRNA-142 levels, and patients with higher levels of this miRNA had longer progression-free median survival.

A single study found a correlation between high miRNA-142 levels and chemo-resistance. Wu et al. [143] compared the miRNA expression profiles of bromocriptine-sensitive and resistant prolactinomas and found 12 miRNAs which were differentially expressed between the two groups. They identified miRNA-142-3p, in addition to miRNA-486-5p as mentioned earlier, as upregulated in the drug-resistant tumors, but no further work was done to establish a mechanism.

Thus, in the case of miRNA-142, the increased circulating levels in obese individuals may contribute to a full response to treatment, particularly for pancreatic cancer, HCC, AML, NSCLC, and ovarian cancer.

### 3.5. Circulating Levels of MiRNA-520c-3p Are Decreased in Obesity

The miRNA-520c gene is located on chromosome 19, it encodes an 87-base precursor miRNA and ultimately results in two mature miRNAs; hsa-miRNA520c-3p and hsa-miRNA-520c-5p. Hsa-miRNA520c-3p (referred to as miRNA-520c here on in) has been shown to demonstrate oncogenic properties in a number of studies [144,145], though whether it promotes or suppresses tumor progression is malignancy dependent. This is further complicated by the specific roles attributed to subtle variations in the miRNA-520 subtypes. Circulating levels of miRNA-520c were found to be decreased in obesity in this review [71,72].

#### MiRNA-520c Is Lower in Hepatocellular Carcinoma and May Affect Resistance to 5-Fluorouracil

MiRNA-520c is an intermediate regulator of TARDBP-mediated regulation of glycolysis in HCC cells and as metabolic changes in metabolism of cancer cells and are often linked with the clinical outcome of patients, the potential role of miRNA-520c is potentially interesting [146]. The expression of miRNA-520c has been observed to be significantly lower in HCC tissues and cells compared with tumor-adjacent tissues and L02 cells and is correlated with poor patient prognosis [145]. In this study, induced overexpression inhibited HCC cell proliferation, migration and invasion, and promoted apoptosis. Changes to 5-FU chemoresistance have also been observed via upregulation of the transcription factor E2F-1 and Thymidylate synthase [147]. E2F-1 has been associated with enhanced chemosensitivity in other cancers due to its activation of Akt [148]. Therefore, in the context of obesity lower levels of circulating miRNA-520c may promote poorer outcomes to HCC treatment.

### 3.6. Circulating Levels of MiRNA-140-5p Are Increased in Obesity

MiRNA-140-5p has been shown to be increased in the circulation of obese patients [71,72]. Although not extensively studied, miRNA-140-5p has also been associated with the migration, invasion and proliferation of a number of cancers including those of the colon, breast, cervical, gastric, and lung. Further to this, its role as a salivary biomarker for parotid gland tumors has been tested [149]. The findings of expression levels in cancer is somewhat contradictory to the obesity-cancer paradox.

#### 3.6.1. Induced MiRNA-140-5p Levels in Breast Cancer May Enhance Outcomes to Doxorubicin Treatment

Expression of miRNA-140-5p is frequently down regulated in breast cancer stem cells and its inhibitory effects on proliferation have been linked to Wnt1 [150]. Mimics of miRNA-140-5p enhanced sensitivity of doxorubicin treatment via the Wnt1/ABCB1 pathway both in vitro and in vivo. Therefore, these findings would suggest increased miRNA-140-5p expression levels may benefit patient response to doxorubicin treatment and reduce chemoresistance [150].

#### 3.6.2. MiRNA-140-5p Influences Treatment Sensitivity in Osteosarcoma

The expression of miRNA-140-5p has been associated with chemosensitivity in osteosarcoma. Experimentally induced knockdown increased multidrug resistance in both in vitro and in vivo systems. The study by Meng et al. [151] identified autophagic pathways involving HMGN5 as pivotal for this change in sensitivity. Further to this, miRNA-140-5p elicits G1 and G2 arrest via induction of p53 and p21, the authors of this study were able to partially sensitise resistant osteosarcoma cells to 5-FU and hypothesized that this was due to inhibition of proliferation [152]. In contrast, chemotherapy has been shown to induce expression of miRNA-140-5p and is associated with increased autophagy via inositol triphosphate signaling, thereby contributing to resistance to doxorubicin and cisplatin [153].

#### 3.6.3. MiRNA-140 Levels Correlate to Chemosensitivity of NSCLC

Levels were decreased in NSCLC tissue compared with matched normal tissues and this expression was significantly associated with N or M classification of patients [154]. Low levels of miRNA-140 correlated with increased sensitivity to etoposide and topotecan in cell lines [155]. Although it has been proposed that it may serve as a tumor suppressive role via VEGF, forced overexpression of miRNA-140-5p in NSCLC cells reduced migration suggesting enhanced sensitivity to gefitinib, DMH1, and cisplatin [156]. Further studies will clarify the role of miR-140-5p in the response of NSCLC to treatment.

#### 3.6.4. Elevated Levels of Circulating MiRNA-140-5p May Promote Enhanced Sensitivity to Treatment in Multiple Myeloma

Increased levels of miRNA-140-5p, achieved using mimics, was associated with suppressed autophagy and enhanced sensitivity to melphalan in resistant multiple myeloma cells [157]. Linc00515 was an upstream regulator of miR-140-5p in this context, and changes in chemo-sensitivity were attributed to changes in ATG14 level, a miR-140-5p target which influences membrane tethering and autophagosome fusion and is therefore implicated in cytoprotective autophagy [158].

#### 3.6.5. MiRNA-140-5p Is Decreased in Gastric Cancer and May Affect Chemosensitivity via SOX4

The expression of miRNA-140-5p in gastric cancer is somewhat complex; high plasma miRNA-140-5p levels have been found in patients, but levels in a small cohort of tissue samples are low, similar to levels in the HGC-27 cell line [159]. Downregulated tissue expression of miRNA-140-5p has been associated with a reduced mean survival of patients [160]. MiRNA-140 overexpression enabled increased cell death of HGC-27 cells after doxorubicin treatment [159], via SOX4 which has been associated with chemosensitivity in other cancers.

In conclusion, increased circulating levels of miRNA-140-5p in obese individuals may contribute to a more complete response to treatment in the case of gastric cancer, MM, osteosarcoma and breast cancer. In NSCLC the predicted phenotypic effect of increased circulating miR-140-5p levels is not clear.

### 3.7. Circulating Levels of MiRNA-143 Are Increased in Obesity

The role of miRNA-143 within the adipose tissue is conflicting, some have identified it as adipogenic [161], whilst others have described it as anti-adipogenic [67]. A study by Chen and colleagues have attributed these differences to the stage of adipocyte differentiation [162]. A further role has been identified in insulin resistance through targeting of the insulin-like growth factor 2 receptor and subsequent activation of the insulin signaling pathway [163]. The expression of miRNA-143 differs in different tissues, its expression is downregulated in subcutaneous adipose tissue of obese individuals [67], whereas animal models show up-regulation of adipose tissue expression, and also it is increased in the liver, pancreas and circulation [164] making its role in cancer and treatment sensitivity complex. MiRNA-143 targets a number of tumorigenic genes including c-Myc, ERG and Ets-1 and is classed as a tumor suppressor [165]. MiR-143 deregulation has been reported as miRNA-143 is most abundantly expressed in colonic tissue and it is widely accepted that there is a loss of miRNA-143 during the transformation of a normal epithelial cell to its malignant state [166]. Both tissue and circulating levels are found at a reduced level compared to normal colonic mucosa [167] although the precise cell type expressing miRNA-143 with functional significance is debated [166]. In stage II CRC patients, low expression of miRNA-143 was significantly associated with better progression free survival and overall survival [168]. Further to this down-regulated miRNA-143 in serum has been observed in colorectal, osteosarcoma, breast cancer and esophageal squamous cell carcinoma [169] and may be indicative of prognosis.

#### 3.7.1. Increased Levels of MiRNA-143 Enhance Chemosensitivity Colorectal Cancer

CRC patients with low expression of miRNA-143 in their primary tumor demonstrated increased survival rates compared with high expression [170]. Further to this, overexpression of miRNA-143 enhanced chemosensitivity to oxaliplatin treatment in an IGF-IR dependent manner [171] and increased sensitivity to 5-fluorouracil [172]. This miRNA can be expressed as a cluster with miRNA-145 and enhanced levels of cetuximab-mediated antibody-dependent cellular cytotoxicity in human CRC cells were associated with overexpression of either miRNA-143 or miRNA-145 [173]. Chemoresistance in colon cancer has also been linked with FXYD3, an ion transport regulator and target of miRNA-143, and an association has been shown with progression free survival [170].

#### 3.7.2. Increased Levels of MiRNA-143 Enhance Chemosensitivity in Bladder Cancer

Serum levels of miRNA-143 in bladder cancer correlate with clinical stage, lymph node metastasis, distant metastasis, and prognosis Despite high levels of miRNA-143 correlating with poor progression-free survival in aggressive bladder cancer [174] this miRNA has been shown to enhance sensitivity to gemcitabine in 5637 cells; this was proposed to be via repression of IGF-1R signaling Further to this, in vitro experiments which induced FOXD2-AS1 knockdown, a long non-coding RNA which can inhibit miRNA-143, enhanced sensitivity to gemcitabine [175].

#### 3.7.3. Increased Levels of MiRNA-143 Enhance Chemosensitivity in Prostate Cancer

MiRNA-143 enhanced chemosensitivity towards docetaxel in prostate cancer [126]. DU145 and PC3 cell lines transfected with miRNA-143 displayed decreased proliferation and migration and enhanced sensitivity to docetaxel through suppression of KRAS, a key molecule of EGFR/RAS/MAPK signaling. KRAS has been implicated in a poorer response to chemotherapy in several clinical trials [176].

#### 3.7.4. Overexpression of MiRNA-143 Attenuates Autophagy in Non-Small Cell Lung Cancer to Enhance Chemosensitivity

MiRNA-143 is down-regulated in NSCLC tissues and cell lines and forced overexpression of this miRNA was able to suppress proliferation, increase apoptosis, and inhibit migration and invasion in NSCLC cell via inhibition of Limk1 [177]. Down-regulation of Limk in NSCLC cell lines enhanced sensitivity of 801D cells to chemotherapeutic drugs of cisplatin and gemcitabine. Furthermore, miRNA-143 has been found to target autophagy-related 2B in non-small cell lung cancer H1299 cells, and as higher sensitivity to 5-FU is observed when the autophagic response is attenuated in A549 cells, it would be likely that changes in autophagy pathways contributes to the enhanced chemosensitivity However, evidence of its role in chemosensitivity appears cancer or strand dependent, as miRNA-143-3p was found to be increased not decreased in exosomes of gemcitabine-resistant NSCLC cells.

#### 3.7.5. Reduced MiRNA-143 Contributes to Breast Cancer Drug Resistance in a Sub-Group Dependent Manner

MiRNA-143 is frequently downregulated breast carcinoma tissues compared with non-cancerous breast tissue [178] and the miRNA-145/143 gene cluster is more greatly expressed in luminal-A than in basal-like breast cancer subtypes [179]. An inverse association between miRNA-143 and CIAPIN1 protein expression levels has been observed in a number of breast cancer cell lines. Overexpression of CIAPIN1 has been associated with multidrug resistance, and this was supported in the study by Wang and colleagues [180]. Furthermore, studies have shown that the miRNA-145/143 cluster targets the 3′-UTR of the ERBB2 gene, which has been linked with breast cancer multidrug resistance in a subtype dependent manner [181].

#### 3.7.6. MiRNA-143 Contributes to Chemoresistance in Osteosarcoma Tumor Cells through Changes in Autophagy Pathways

In osteosarcoma, miRNA-143 expression is significantly downregulated in both tissues and cell lines [182] and this loss of expression is correlated with shorter survival of patients with osteosarcomas underlying chemotherapy [183]. MiRNA-143 levels were also found to be downregulated in chemo-resistant SAOS-2 and U2OS osteosarcoma cells, leading to changes in autophagic pathways, namely via ATG2B, Bcl-2, and/or LC3-II.

#### 3.7.7. MiRNA-143 Is Associated with Chemoresistance Mechanisms in Cervical Cancer

There is little evidence of miRNA-143 changing chemosensitivity in cervical cancer despite increasing down-regulation of miRNA-143 being observed in increasing histology severity [184]. Taxol therapy of patients with cervical squamous cell cancer demonstrated an increase expression of miRNA-143 but this was not linked with changes in sensitivity to treatment [185]. Furthermore, it has been shown that expression of E3 ubiquitin ligase isolated by Differential Display (EDD) may promote cervical cancer growth in vivo and in vitro by targeting miRNA-143 [186]. EDD has been demonstrated to a role in drug resistance and therefore may prove to be a link between miR-143 and drug resistance in this cancer.

To summarize, increased circulating levels of miRNA-143 in obese individuals may contribute to a more complete response to treatment in the case of CRC, bladder cancer, prostate cancer, NCLSC, osteosarcoma and some types of breast cancer and are not likely to be an underlying cause of poorer treatment responses.

### 3.8. Conclusions

Our literature review identified 6 key miRNAs which were robustly increased in the circulation of obese patients: miRNAs-486, -142-3p, -222, 140-5p, -122, and -143. These have been linked to a diverse range of cancers and their response to therapy. Most of the identified miRNAs were linked to HCC, which may reflect the central role that the liver and adipose tissue play in whole body energy metabolism [187]. Many of the studies identified clusters of miRNAs that were changed in obesity. Therefore, stringent mapping of cell signaling mechanisms may identify useful downstream targets which could produce significant improvements in cancer outcome, as opposed to targeting just one miRNA which may be ineffective in vivo. All of the miRNAs identified here were detected in the circulation and identifying the source tissue may also yield interesting avenues of enquiry.

At present the data from human patients is limited and larger in vivo studies are required to validate the significance of the findings shown. In our analysis, we have clustered together studies to identify meaningful trends, however, it is likely that the contribution of individual miRNAs in the response to chemotherapy is probably more nuanced, for example the categorization of obese versus morbidly obese may also differentially affect interpretation of the response. Also, different age groups may have different responses; a recent systematic review by Oses and colleagues reports increased circulating levels of several of the miRNAs reported here, and by Zaiou and colleagues [96], specifically in the pediatric context [120]. In addition, we have reviewed the underlying mechanisms mediated by these key miRNAs which may contribute to treatment responses at the cellular level and could offer novel insights to improve obese patient outcomes.

The work presented here also identifies 3 circulating miRNAs which may be useful for predicting treatment success in obese individuals. High levels of miRNA-486 may be a biomarker for poorer treatment outcomes in obese children with CML. Increased miR-222 levels are already linked to poorer outcomes in response to treatment for cancer and, when combined with the information reviewed here, may be useful biomarkers of poorer prognosis in breast and bladder cancers, NSCLC and MM in obese patients. Finally, lower circulating levels of miRNA-520c may predict poorer outcomes for obese patients treated for HCC. Clearly further work is required to validate these proposed trends and determine any use for patient stratification. To conclude, circulating miRNAs have the potential to direct appropriate personalized care, or predict prognosis, but much more work is necessary for validation of these as tools in the clinical setting.

## 4. Materials and Methods

The literature review was performed in two stages. Stage 1 aimed to identify which circulating miRNAs are up or down-regulated in obese individuals compared with typical weight individuals. PubMed was searched using the terms “obesity, obese, microRNA, miRNA, circulating, blood, serum, plasma” in various combinations to identify relevant studies. Original articles and reviews were considered. There were no language restrictions at this stage, and only studies performed in humans were considered. Importantly, the studies identified did not include cohorts of cancer patients, so any observed changes were due to obesity and not cancer-associated. The literature search was performed in January 2019. The remit of this review does not include a comprehensive discussion of the role of individual miRNAs in the development of cancer or obesity (reviewed [161]), but instead is focused on the role of relevant miRNAs in the response to chemotherapy at the cellular level. As this review was focused on identifying circulating miRNAs which are altered in obesity, articles relating to altered miRNAs identified within adipose tissue, as opposed to those identified in the circulation, were excluded at this stage. Abstracts were reviewed for relevant content and relevant miRNAs identified and listed in Table A1 (Appendix A). All studies describing circulating extracellular miRNAs were considered and no exclusions were made based on the source e.g. exosomes, blood, plasma or serum, or the type of patient, e.g. paediatric or adult. In order to identify robust changes miRNAs which were identified to be both up and down-regulated in conflicting studies are shown in Table A1 in underlined text and were excluded from stage 2. MiRNAs which were identified to be regulated in the same direction in more than 1 study were taken forward to stage 2.

Stage 2 consisted of a systematic literature review to identify relevant literature regarding the role of the 8 candidate miRNAs in mediating sensitivity to cancer treatment. PubMed was searched using the terms “miRNA-XXX, microRNA-XXX, cancer, drug resistance, drug sensitivity” in various combinations to identify relevant studies. Original articles only were considered, so the primary data could be evaluated. The literature search was performed in March 2019. One article was excluded as it was not available in English, another was excluded due to subscription restrictions.

## Figures and Tables

**Table 1 ncrna-06-00005-t001:** Circulating microRNAs (miRNAs) which were reproducibly increased in obesity.

MiRNA	Source (Number of Patients)	Reference
486	Pediatric plasma (85 control, 40 obese)	[68]
	Pediatric plasma (156 control, 100 obese)	[69]
	Pediatric serum (12 control, 17 obese)	[70]
142-3p	Plasma from morbidly obese males (12 control, 8 morbidly obese)	[71]
	Plasma from adolescents (50 control, 100 obese)	[72]
222	Plasma from morbidly obese males (12 control, 8 morbidly obese)	[71]
	Pediatric plasma (156 control, 100 obese)	[69]
	Plasma from adolescents (50 control, 100 obese)	[72]
	Pediatric samples (10 control, 20 obese)	[73]
	Paediatric plasma (85 control, 40 obese)	[68]
140-5p	Plasma from morbidly obese males (12 control, 8 morbidly obese)	[71]
	Plasma from adolescents (50 control, 100 obese)	[72]
122	Serum from male young adults (107 control, 123 obese)	[74]
	Pediatric serum (12 control, 17 obese)	[70]
	Pediatric samples (10 control, 20 obese)	[73]
	Fasting venous blood (20 control, 30 obese)	[75]
143	Fasting venous blood (20 control, 30 obese)	[75]
	Plasma from adolescents (50 control, 100 obese)	[72]

**Table 2 ncrna-06-00005-t002:** Circulating miRNAs which were reproducibly decreased in obesity.

MiRNA	Source (Number of Patients)	Reference
221	Pediatric plasma (85 control, 40 obese)	[68]
	Plasma from morbidly obese males (12 control, 8 morbidly obese)	[71]
520c-3p	Plasma from morbidly obese males (12 control, 8 morbidly obese)	[71]
	Plasma from adolescents (50 control, 100 obese)	[72]

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
