# Peer review of "MiRNAs as Novel Adipokines: Obesity-Related Circulating MiRNAs Influence Chemosensitivity in Cancer Patients"

_ncrna, 2020, doi:10.3390/ncrna6010005_

Round 1
Reviewer 1 Report
I appreciate the opportunity to review this interesting review on the circulating adipose-derived microRNAs in cancer patients. In my opinion the manuscript is interesting and original and of clinical relevance. Also it is well written and data are appropriately presented. To my knowledge, no similar review was published recently and the review contains all major and important papers in this field. In my opinion the paper is publishable in its current version. However some added tables or figures would help the reader to understand the topic better.
Author Response
We thank the reviewer for their positive comments.

Reviewer 2 Report
This is an interesting and complete review about the potential role of microRNAs as adipokines that exert distant effect in different tissues. In particular, this review is focused in the role of adipokines in the response of tumors to chemotherapy in the hallmark of obesity.
Minor concerns:
1) Further relevant information such as population size, upregulated and downregulated miRNAs (tissue and/or secreted) should be included in the Tables. Tables should be further detailed in order to amplify the information.
2) The role of miRNAs as adipokines is not completely clarified and this should be mentioned throughout the text.
3) A paragraph including limitations and new perspectives for this research line should be included at the end of the manuscript.
Author Response
We thank the reviewers for their helpful comments. In response we have made the following improvements to the manuscript which we have also uploaded for you to look at:
We have included new information in tables 1 and 2 to indicate the population size used for each cited study (see the 'Source' column). Th results are presented as 2 results tables, one for upregulated miRs (table 1) and one for downregulated miRs (table 2). We have not included information about the likely tissue of origin of these miRs, as in most cases this would require an extensive amount of further literature review to determine, and we believe this is out of the scope of this piece of work. Certainly it is an interesting next step for this study (now referred to in lines 717-719).
We have emphasized that the role of miRs as adipokines in novel and still not clarified (see lines 104-106 and lines 154-156).
We have included an extra paragraph in the Conclusions section to further highlight some new perspectives arising from this review (see lines 730-737). We have also further considered the limitations of our review (lines 163-166) alongside our previous observations (lines 720-729).
